# Genome-Based Taxa Delimitation (GBTD): A New Approach

Alexander Bolbat, Yurij Bukin and Irina Kaygorodova *

Limnological Institute SB RAS, Irkutsk 664033, Russia
* Correspondence: irina@lin.irk.ru

**Abstract:** Proper taxonomic identification is essential for biological research. Unfortunately, there are no clear guidelines for taxonomic assignment above the species level. Here, we present a novel approach—GBTD—to the use of genetic divergence to evaluate the taxonomic position of certain samples with simultaneous estimation of the current systematics correctness. This approach includes measuring the raw and model-adjusted distances between DNA sequences and attributing them to the lowest taxonomic levels that are common in sample pairs to reveal distance distributions matching different taxonomic levels (species, genus, family etc.). GBTD facilitated the reassessment of the taxonomic position of the samples, whose genetic distances relative to other samples in the dataset did not match their taxonomic divergence. A data set of complete mitochondrial genome sequences of segmented worms was chosen to test this approach. As a result, numerous inconsistencies in the systematics of samples from GenBank were pointed out. These inconsistencies included both the oversplitting and overlumping of individuals into taxa of different levels and clear cases of misidentification. Our approach sparks re-evaluation of the current systematics where traditional methods fail to provide sufficient resolution.

**Keywords:** genetic divergence; high-level taxa delimitation; molecular-based systematics; oligochaetes; leeches; leech-like parasites

## 1. Introduction

The consequences of improper taxonomic assignments reach far beyond mislabeled samples. Both uniting evolutionary distant organisms into the same taxon and splitting closely related organisms into distant taxa can impede any downstream analysis. Imbalanced datasets consisting of overrepresented and underrepresented taxa may lead to the distortion of effects in ecological modeling and branch lengths in phylogenetic research. Unfortunately, despite centuries of dispute and controversy, the guidelines for assigning organisms to certain taxonomic levels have not been established. Admittedly, all human attempts to systematize living organisms are artificial, and there are no strict cut-off boundaries between taxa in nature. Phenomena such as cryptic diversity, introgression and incomplete lineage sorting blur our very concept of even the most basic taxonomic level–the species–which seems to have been studied backwards and forwards. The criteria for including species into higher-level taxa are even more arbitrary. There are no recommendations on which degree of divergence of which characters should serve as a basis for considering species as members of the same genus, family, order, etc.

Uniting evolutionarily distant organisms into the same taxon may result in misguided estimates of studied effects, while splitting closely related organisms into distant taxa may cause their unjust exclusion from the dataset.

Ideally, decisions about taxonomic assignment should come from the information contained within samples. However, morphological criteria alone do not provide sufficient resolution for modern research, which leads to a need to use more resolved genetic data. The most successful project using genetic information to solve problems in taxonomic assignment started in 2003 and was called DNA barcoding [1]. During the days of its inception, an arbitrary 2–3% threshold value of genetic distance was established to indicate

the maximum intraspecies divergence value for the *cox1* fragment [2]. This threshold raised the suspicion that it might be taxon-specific, so newer approaches capable of leveraging divergence times [3], branching patterns [4], and genetic distance frequencies [5] were developed. These methods were successfully applied to different groups, e.g., leeches [6], gastropods [7] and copepods [8]. However, all these approaches are confined to a single taxonomic level, i.e., species, neglecting the rest of the hierarchic levels of taxonomy. Meanwhile, taxonomic identification at a higher level than species level can suffer from the same issue of lacking distinct characters for delimitation. We believe that the principle of a "DNA barcode gap" can be extended to higher taxonomic levels. Our approach suggests measuring which genetic distances and divergence times match certain taxonomic levels with simultaneous re-evaluation of current taxonomy and extrapolating these values to the organisms of interest.

Although the technique of DNA barcoding was initially developed with the *cox1* fragment in mind, this marker may not be suitable to resolve higher evolutionary relationships [9]. Animal mitochondrial genomes are widely used in evolutionary studies and population genetics. Due to their generally uniparental inheritance and limited recombination, they are often thought to be ideal markers, and their length and nearly ubiquitous presence in eukaryotes allows reproducible results in evolutionary studies. Although some convincing arguments were made stating that mitochondrial genomes are far from ideal markers [10], they, nevertheless, remain a valuable source of information. The arguments in favor of using mitochondrial markers in evolutionary and taxonomic research remain unchallenged and no single nuclear marker has been found which can reliably outperform them [11]. The abundance of complete mitochondrial genomes enables their use as standardized markers for phylogeny reconstruction and taxonomic research, unlike numerous loci that can be produced using the enrichment approach [9]. A typical metazoan mitochondrial genome has a conservative set of orthologous genes, and a length of approximately 16.5 kb, which is sufficient for resolving ancient phylogenetic relationships [12].

For approbation of the new experimental approach to the complex problem of distinguishing between taxa of different levels, representatives of the order Acanthobdellida Grube, 1850 were chosen, as a simple and understandable object. This group consists only of two species: *Acanthobdella peledina* Grube, 1850 and *Paracanthobdella livanowi* (Epstein, 1966) [13]. It is assumed to be an evolutionary link between oligochaetes and leeches [14]. The range of *A. peledina* spans from Scandinavia to Chukotka [15] and probably Alaska [16,17], whereas the distribution of *P. livanowi* is limited by the fresh waters of Kamchatka and Chukotka [15,18]. The logical question concerning *A. peledina* of whether this entire vast area is inhabited by a single species still remains unanswered. Another issue concerns the attribution of the species *P. livanowi* as a single member of the genus *Paracanthobdella* in the separate family Paracanthobdellidae [19]. This formation of a new family for a single species was not motivated by Epstein [19], but subsequently, was never disputed by anyone. We will try to figure out this issue regardless.

## 2. Materials and Methods

### 2.1. New Method Description

The GBTD approach relies on a simultaneous analysis of samples' taxonomy, raw genetic distances and branch lengths in phylogenetic trees. The branch lengths in phylogenetic trees are supposed to represent model-adjusted genetic distances; thereby, simple cladograms with no meaningful branch lengths would not be useful. Multiple samples of different taxonomic levels are essential in phylogenetic analysis to generate a great number of pairwise comparisons, infer their distributions and recursively correct taxonomy based on these distributions.

Pairwise distance calculation is performed in the R [20] script that uses the dist.dna and cophenetic methods of the ape package [21] and four input files (Figure 1). The nucleotide alignment file in the Fasta format is used to calculate a matrix of raw pairwise genetic distances as proportions of sites differing between individuals (p-distance). The phyloge-

netic tree can be imported in the Newick format. Both ultrametric and non-ultrametric phylograms can be used, although the ultrametric tree is more convenient for spotting divergence times that do not quite match the taxonomic divergence. The third file is a comma-separated value (CSV) table containing all the information about the sample taxonomy. The first column contains the sequence ID, the same as in nucleotide alignment and tree files, and the other columns contain sample taxonomy from higher to lower levels. The GBTD script measures pairwise distances by calculating the p-distance and the sum of tree branch lengths between individuals, which represent model-adjusted distances. Then, the script iterates through the columns of the taxonomy table to find the lowest taxonomic level shared by two individuals. Once the lowest shared taxonomic level is found, distance values between individuals are assigned that taxonomic level. Then, these annotated distances are plotted in three output files. Finally, if the distances in some specific sample pairs are inferred, these pairs are imported from the fourth file in the CSV format and depicted as squares on all output plots. The GBTD script also supports sample exclusion by passing a vector of sample IDs as one of the arguments.

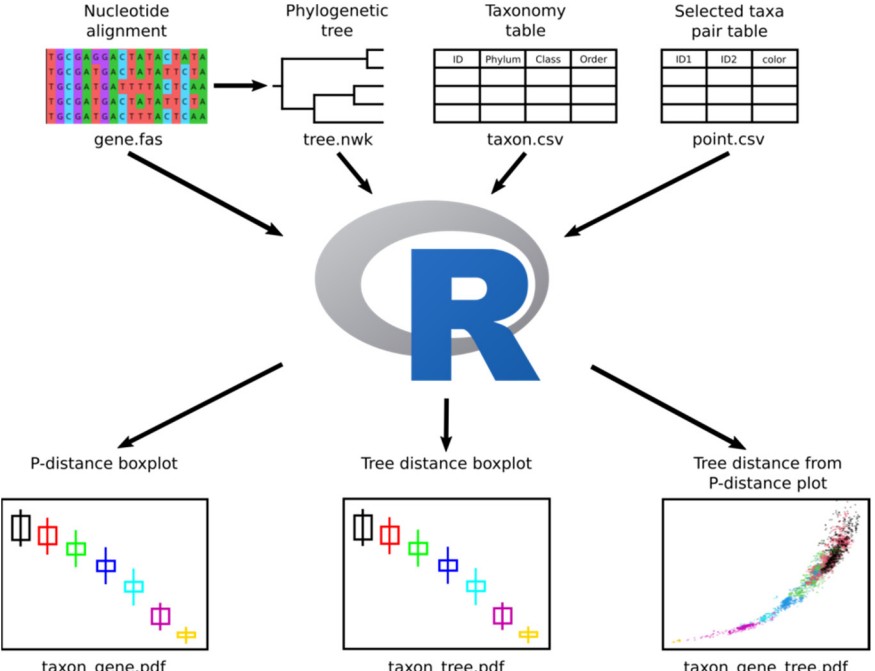

**Figure 1.** Schematic flowchart of the experimental approach to taxonomic delimitation through genomic data. Input files (top of the image) are nucleotide alignment in Fasta format, phylogenetic tree in Newick format and CSV tables with taxonomy information and selected specimen pairs. Output images (bottom of the image) of the proposed script are named based on input file names.

The GBTD script and all input files used for this work are available on the FigShare platform (https://figshare.com/s/00a21013944f83039a43, accessed on 1 November 2022).

### 2.2. Sample Collection, DNA Extraction and Sequencing

To evaluate the new approach in differing taxonomic scopes, an experiment framework was designed (Figure 1). We selected various representatives of freshwater Clitellata with well-characterized mitochondrial genome data (Tables S1 and S2—Supplemental data) as well as samples that were available for re-assembly and newly sequenced (Table 1).

**Table 1.** List of the Clitellata mitochondrial genomes obtained in this study, with their sampling locations (for newly sequenced samples), Sequence Read Archive accession numbers (for re-assembled samples from public data) and GenBank accession numbers.

| Species | Sampling Location and Coordinates or Data Accession | GenBank Number |
|---|---|---|
| *Acanthobdella peledina* | River Pitea, Sweden; 66.449778, 18.03977 | MZ562997 |
| | River Iya, Irkutsk Region, Russia; 53.804354, 99.558638 | OM203184 |
| | River Middle Rassokha, Irkutsk Region, Russia; 58.001986, 109.466141 | OM203186 |
| | Lake Bolshoy Kharbey, Nenets AD, Russia; 67.556111, 62.889722 | OM117616 |
| | Lake Labynkyr, Sakha (Yakutia), Russia; 62.498315, 143.598658 | OM203185 |
| | River Yana, Sakha (Yakutia), Russia; 67.553945, 133.367242 | OM214536 |
| *Paracanthobdella livanowi* | Lake Azhabachje, Kamchatka, Russia; 56.146114, 161.802687 | OM117614 |
| | | OM117615 |
| *Codonobdella* sp. | Lake Baikal, Russia; 53.240811, 107.234195 | MZ202177 |
| *Baicaloclepsis grubei* | Lake Baikal, Russia; 53.240811, 107.234195 | OM257165 |
| *Baicaloclepsis echinulata* | Lake Baikal, Russia; 53.012873, 106.920606 | OM257166 |
| *Erpobdella octoculata* | SRA accession: SRX9009198 | OM257408 |
| *Glossiphonia complanata* | SRA accession: SRX8928147 | OM039422 |
| *Haemopis sanguisuga* | SRA accession: SRX9009141 | OM234778 |
| *Haemopis sanguisuga* | SRA accession: SRX9009400 | OM234779 |
| *Lumbriculus variegatus* | SRA accession: SRX9009164 | OM062609 |
| *Piscicola geometra* | SRA accession: SRX9009199 | BK059172 |
| *Theromyzon tessulatum* | SRA accession: SRX8928146 | OM039423 |

Acanthobdellidans represented by two species, *Acanthobdella peledina* and *Paracanthobdella livanowi*, were identified as the main object for testing the new approach. Their samples were collected directly from an appropriate host caught in a vast geographic area in the Palaearctic region: from Scandinavia to Kamchatka (Table 1). Additionally, *Codonobdella* sp., *Baicaloclepsis grubei* and *Baicaloclepsis echinulata* leeches endemic to Lake Baikal were sampled from the lake (Table 1) at a depth of up to 186 m using various pieces of hydrobiological equipment. Taxonomic identification of the collected samples was carried out according to up-to-date keys [15,19].

Total DNA was isolated from the entire organism for all samples, except for the large leech *B. grubei*, from which only the posterior sucker was taken with a sterile scalpel. Additional measures were taken to prevent contamination of the target DNA by the genetic material of the host (fish or mollusk depending on the leech species). Before DNA isolation, individuals were cleaned of mucus and rinsed with 70% ethanol. Then, the samples were dissected, the intestines were removed from them with sterile tools, and they were rinsed again. The cleaned tissues were rehydrated by placing them in distilled water for 10 min. The rehydration procedure was repeated three times, each time with a change of water followed by pipetting. Thereafter, the liquid was removed, and the sample was ground to a homogeneous state using a plastic pestle. DNA extraction from the homogenate was performed using the DiaGene kit (Dia-M, Cat. No. 3319.0250) based on ion-exchange columns. The precipitate was washed with 80% ethanol, dried and dissolved in 50 µL deionized water.

The purified solution of total genomic DNA was then sheared to 300-bp fragments using the Covaris M200 sonicator. The NGS libraries were then prepared with the NEBNext Ultra II DNA Library Prep Kit (New England Biolabs, Cat. No. E7645L) and sequenced with the NextSeq 550 Mid Output Kit v2.5 (Illumina, Cat. No. 20024904) at the IC&G Center of Genomic Investigations (Novosibirsk, Russia).

*2.3. Genome Assembly and Annotation*

Quality control of the sequencing output was performed using FastQC [22]. Adapter clipping and quality trimming were carried out in Trimmomatic v. 0.40 [23]. Contig assembly was performed in Mira v. 5 [24]. The contigs were visualized in the Tablet genome browser [25]. Mitochondrial contigs were identified using the BLASTn homology search [26] performed on highly covered AT-rich contigs.

De novo mitochondrion genome (mitogenome) annotation was performed in three stages. First, the mtRNA genes were annotated with Aragorn v 1.2.40 [27]. Overlapping tRNA genes were annotated according to the tRNA-punctuation theory: with the intact 5' end of the downstream gene and the truncated 3' end of the upstream gene [28]. At the second stage, protein-coding genes were annotated via a homology search for amino acid sequences of open reading frames in BLASTp [29].

According to the tRNA-punctuation theory, protein-coding genes can overlap with each other but cannot overlap with the tRNA genes. At the final stage, the rRNA genes were annotated using barrnap [30].

*2.4. Mitogenome Alignment and Phylogeny Reconstruction*

Newly sequenced data were supplemented with the Clitellata mitochondrion genomes available in GenBank that matched the following criteria: length of at least 13 kb, a full set of annotated mitochondrial genes and no duplication in the database (in particular, out of 72 sequences available in GenBank for Oligochaeta, 26 were omitted as duplicates). Precise taxonomic identification was not considered a criterion. Thus, the data obtained in this study were supplemented by 69 well-characterized sequences from Gen-Bank (Tables S1 and S2—Supplemental data).

Additionally, to enrich the comparison group for further analysis seven mitochondrial genomes were reassembled from raw data previously published in the SRA database (Table 1).

Due to multiple rearrangements in the mitochondrial genomes, the orthologous genes in the dataset had to be placed in a uniform order prior to alignment. This task had to be carried out manually according to publishers' annotations since existing software packages for rearranged genomic sequences turned out to be not sensitive enough for distantly related genomes. After this manual rearrangement, the sequences were aligned with Mafft v 7.453 [31]. For both Maximum Likelihood and Bayesian phylogeny reconstructions, the dataset was manually partitioned into three partitions. The first partition included codon positions 1 and 2, the second partition included codon position 3, and the third partition included all non-coding genes. Poorly aligned regions were not deleted from the dataset, but indel sites were not partitioned as meaningful data and, thus, were excluded by the default settings of the phylogenetic software. The GTR+G+I+X substitution model was used for all reconstructions. The Maximum Likelihood phylogeny was reconstructed using IQ-TREE v2.1.3 [32] with topology support estimated by ultrafast bootstrap approximation [33]. The Bayesian phylogeny reconstruction was then carried out in BEAST v 2.6.6 [34] using one model for all partitions with a total of 200 million MCMC generations, where every 5000th generation was sampled, and the first 10% of generations were discarded as burn-in. A relaxed lognormal molecular clock parameter was chosen based on the recommendations due to the fact that the 95% highest posterior density of the substitution coefficient rate variation (CV) did not include a zero value [35]. The convergence of ESS statistics was checked using Tracer v 1.7.1 [36].

**3. Results**

To test the GBTD method for detecting high-level taxonomy inconsistencies, we compiled a dataset of 96 complete mitochondrial genome sequences of annelids, including 11 newly sequenced acanthobdelidans and hirudineans (Table 1), seven sequences of different hirudinean and oligochaete species reassembled using raw data from the SRA database (Table 1), and 78 well-characterized mitogenomes of annelids available in Gen-

Bank (Tables S1 and S2—Supplemental data), of which eight polychaete sequences were considered an outgroup. The mitochondrial genomes assembled and downloaded for this study ranged in length from 13,494 to 18,528 bases, with the *A. peledina* genomes being the longest ones. The feature that makes *A. peledina* genomes so long is a repetitive *atp6* pseudogene region between the *atp6* and *trhR* genes, which is absent from all other genomes under study, even from the sister species *P. livanowi*. The number, length and order of repeats in this region are not identical in samples from different geographic locations. The shortest sequence used in this dataset is 13,494 bases long and belongs to *Whitmania acranulata* KM655838.

The reassembly of publicly available raw sequencing data allowed us to extend the length of the respective mitochondrial genomes, in some cases by as much as 7 kb. The SRA and GenBank accessions of the reassembled genomes are listed in Table 1.

The total nucleotide alignment length was 25,835 sites, of which 11,344 sites contained gaps in one or multiple sequences. Part of the total alignment (9.5%) comprised invariant sites.

The first step of our approach consisted of visual estimation of problematic taxonomic assignments in phylogenetic trees. As the lengths of the tree branches can represent model-adjusted genetic distances, their comparison between taxa of similar level may yield useful insight. For taxa of similar level, the branch lengths uniting their members were similar. Therefore, only trees with meaningful branch lengths will be useful regardless of their reconstruction method. Both ultrametric and non-ultrametric phylogenies reconstructed based on a test dataset had nearly identical topologies with minor differences in the placement of closely related samples and the branching of paraphyletic freshwater oligochaetes. Due to the easier visual inspection of ultrametric phylogeny and higher values of node supports, only the ultrametric tree (Figure 2) will further be analyzed with the non-ultrametric data (Figures S1–S3—Supplemental data) used as confirmation and a way to estimate model-adjusted genetic distances.

To estimate the most likely taxonomic position of each individual specimen in the dataset, we applied the approach of measuring both raw and model-corrected (phylogenetic) pairwise distances. For this purpose, we sorted pairwise distances by taxonomic levels common for the respective individuals in the dataset according to their systematics as defined by taxonomy guides and as stated in GenBank. As expected, the values of the distances showed a downward trend from high to low taxonomic levels, albeit with a significant degree of overlap. Specimens of the order Acanthobdellida are currently split into two different families; however, the values of genetic divergence between the species *A. peledina* and *P. livanowi* were grouped significantly lower (15.6% of nucleotide substitutions) than most of the values, characteristic of intra-order level (Figure 3). This suggests that the two species of the order Acanthobdellida are separated by more taxonomic levels than it is necessary in current classification.

Statistical assignment of the Acanthobdellida samples to certain taxonomic levels was hampered by multiple inconsistencies in specimen taxonomic placement compared to their phylogenetic position. In particular, all individual specimens assigned to the genus *Metaphire* are clustered within the clade of the genus *Amynthas* in no discernible pattern. Other authors described similar clustering [37–39], and multiple synonymies were reported between these two genera [40], drawing to the conclusion that all these samples should belong to one genus. The same could be said about *Duplodicodrilus schmardae* KT429015, although in the case of a single specimen, misidentification may also be an explanation. According to phylogenetic analysis, the specimen of *E. octoculata* KC688270 was erroneously assigned to *Erpobdella* and more likely belongs to the genus *Whitmania* (Figures 2 and 3). *Hirudo nipponia* KC667144 clustered separately from other representatives of the genus *Hirudo* into the genus *Whitmania* as *Hirudinaria manillensis* KC688268 did. All of these cases can be explained by sample misidentification.

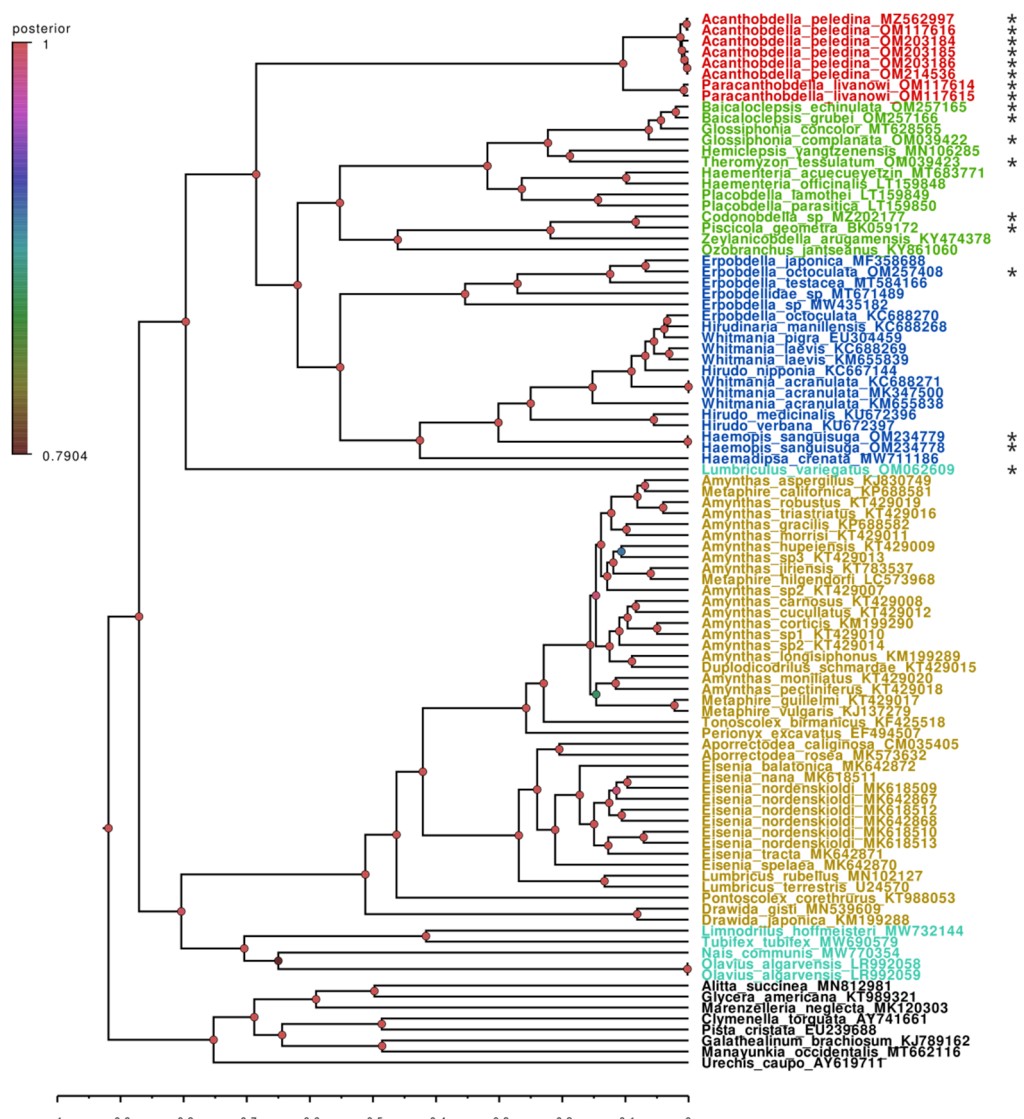

**Figure 2.** Bayesian tree based on complete mitochondrial genome alignment of 96 annelid specimens, including 88 clitellates and 8 polychaetes used as outgroup. Tip labels are colored according to taxonomic and ecological niche: red—order Acanthobdellida, green—order Rhynchobdellida, blue—order Arhynchobdellida, teal—freshwater oligochaetes, brown—soil oligochaetes. Mitochondrial genomes assembled in this work are marked with asterisk (*).

Fixing these issues already improved the statistics; however, more inconsistencies remained. Two genomes of *Whitmania acranulata* (KC688271 and MK347500) are nearly identical in nucleotide composition but have some significant differences from KM655838, resulting in outlier values of intraspecific distances and branch lengths. Furthermore, all *Whitmania* samples are currently assigned to the family Haemopidae, which is split by members of Hirudinidae in phylogenetic trees, making Haemopidae paraphyletic (Figure 2 and Figure S1 of Supplemental data). Based on our analysis, the value of genetic divergence between these families was well within the boundaries of the intra-family range (Figure 3).

Moreover, analysis of the topology of the obtained trees (Figure 2 and Figure S1 of Supplemental data) revealed that the samples of the Baikal glossiphoniid leeches of the genus *Baicaloclepsis* were grouped within the clade of the genus *Glossiphonia*, with a genetic distance of 7.96% between the genera. However, both the raw genetic distances and divergence times of representatives of *Glossiphonia* did not support this split (Figure 3), suggesting the dissolution of the genus *Baicaloclepsis*.

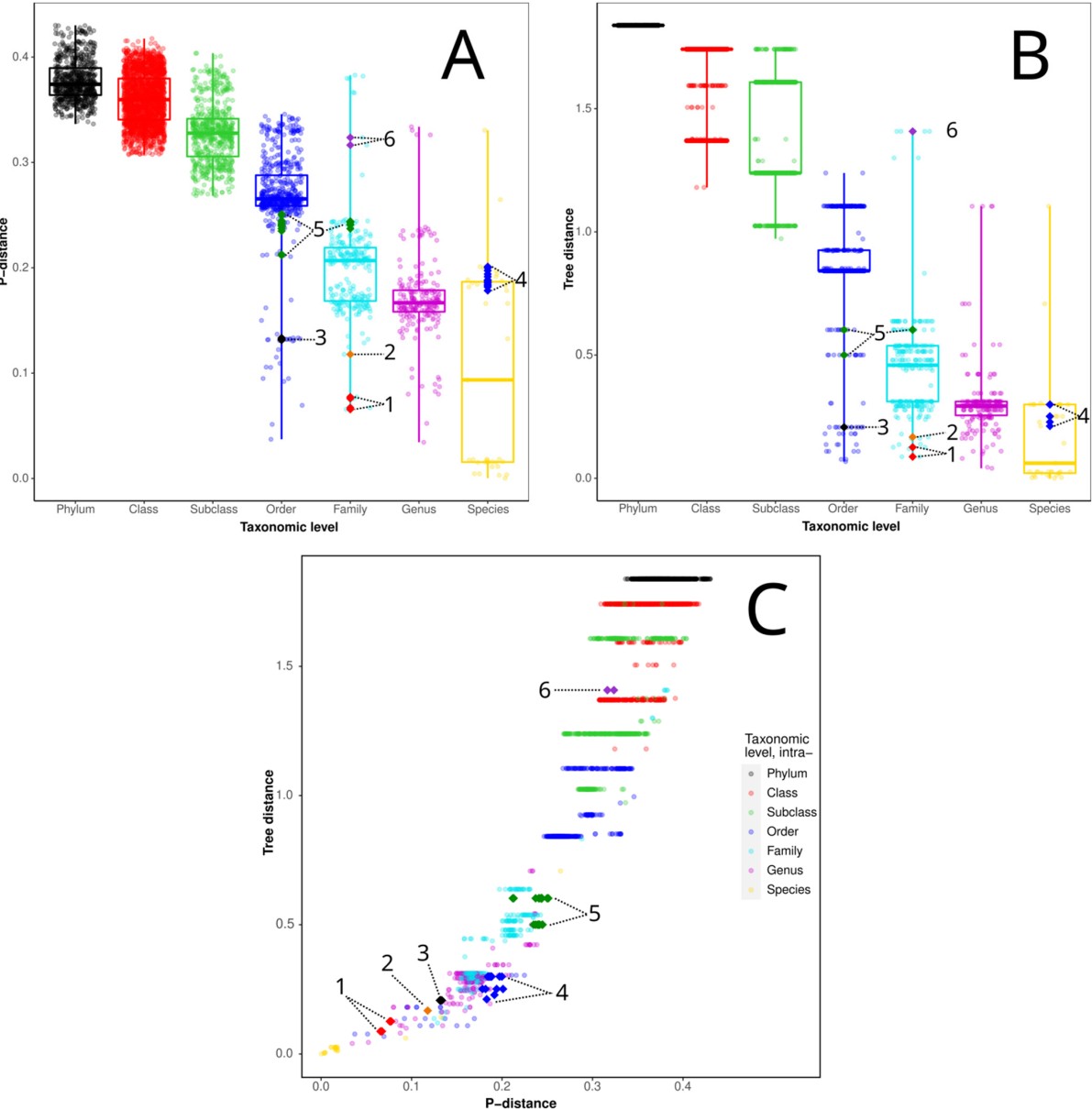

**Figure 3.** Distribution of p-distances (**A**), tree-based distances (**B**), and dependence of ultrametric tree-based distances from p-distances (**C**) by taxonomic levels according to current sample taxonomy. Colored diamonds indicate distances between selected groups: 1—between *Glossiphonia* spp. and *Baicaloclepsis* spp.; 2—between *Piscicola geometra* and *Codonobdella* sp.; 3—between *Acanthobdella peledina* and *Paracanthobdella livanowi*; 4—between samples of *Eisenia nordenskioldi*; 5—between species of families Hirudinidae and Haemopidae; 6—between samples of combined family Naididae (including Tubificidae).

Among piscine leeches, *Codonobdella* sp. is genetically more divergent from *Piscicola geometra* (12.36%) than the species *Baicaloclepsis* is from the species *Glossiphonia*, but its divergence is still within the boundaries of intra-genus level (Figure 3).

Samples of *Eisenia nordenskioldi* have a 20% genetic distance within the group and show divergence times more characteristic of different species within a genus (Figure 3). Freshwater oligochaetes of the family Naididae creates a cluster of outlier family distances above intra-order level, as seen in Figure 3.

Adjusting the samples' taxonomy to fix these issues results in a much clearer downward trend with fewer overlaps of distances (Figure 4). The distance values at intraspecies

level are now clearly distinguishable from the intra-genus values, which indicate the correctness of the DNA-based approach. The distinction between intra-genus and intra-family, as well as between intra-family and intra-order distances is less clear and more noticeable in the divergence times. With progression to higher taxonomic levels the sensitivity of this approach drops and the grouping of divergence times may imply the existence of taxa that have not yet been characterized. However, this grouping occurs due to the small number of nodes at these levels compared to lower levels, and not every node deserves its own taxonomic level.

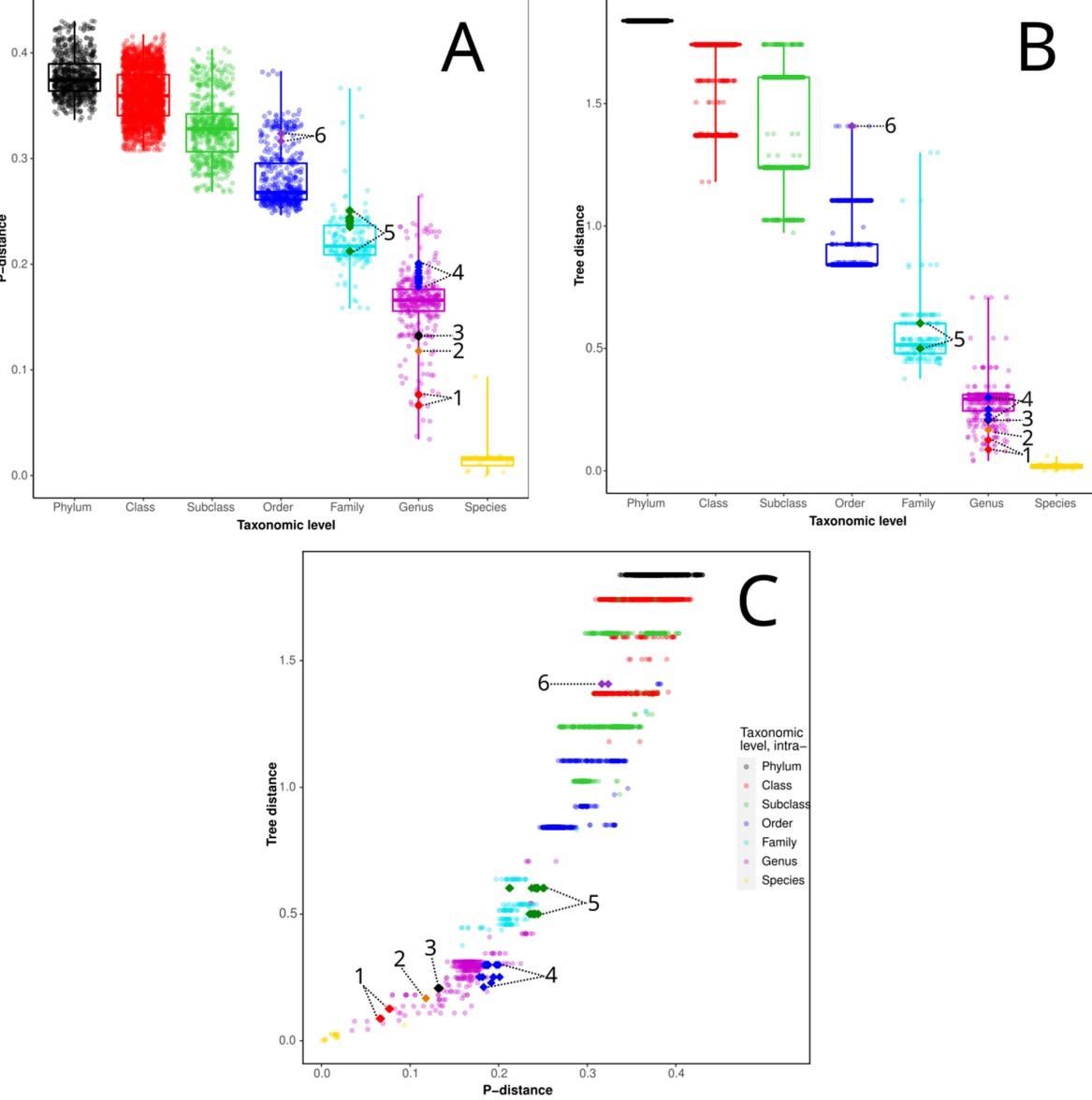

**Figure 4.** Distribution of p-distances (**A**), tree-based distances (**B**), and dependence of ultrametric tree-based distances from p-distances (**C**) by taxonomic levels according to adjusted sample taxonomy. Coloured diamonds indicate distances between selected groups: 1—between *Glossiphonia* spp. and *Baicaloclepsis* spp.; 2—between *Piscicola geometra* and *Codonobdella* sp.; 3—between *Acanthobdella peledina* and *Paracanthobdella livanowi*; 4—between samples of *Eisenia nordenskioldi*; 5—between species of families Hirudinidae and Haemopidae; 6—between samples of combined family Naididae (including Tubificidae).

## 4. Discussion

Both the ultrametric and non-ultrametric trees reconstructed for this study (Figure 2 and Figure S1 of Supplemental data) have topologies close to the modern concept of Annelida evolution. The sequences of *A. peledina* and *P. livanowi* clustered as a sister group of leeches. The group closest to leech-like parasites is freshwater oligochaetes, which corresponds to the hypothesis of an intermediate evolutionary position of acanthobdellids. Sequences of *A. peledina* and *P. livanowi* formed a sister clade to all sequences of the subclass Hirudinea. The divergence times between *A. peledina* samples from different habitats on the ultrametric tree do not reveal any geographic structure within the species, with European samples being only marginally closer to each other than to the Siberian ones, and with the clustering of the latter having no correlation with geography. The closely related species *P. livanowi* clustered separately being more genetically divergent (genetic distance of 15.6% to *A. peledina* sequences). However, both genetic distance and divergence time between two acanthobdelid species are more similar to the values between species of the same genus, as shown by our proposed method (Figures 3 and 4). The species *P. livanowi*, originally described as *A. livanowi* [13], was later transferred into a monospecific genus *Paracanthobdella*, for which a separate family Paracanthobdellidae was established [19]. All these rearrangements were not motivated in any way by V. Epstein [19]; however, they were agreed with in later morphological studies [18,41].

An important result obtained in the phylogeny reconstruction of Clitellata using complete mitochondrial genomes is a monophyly of the order Rhynchobdellida, which was rejected by authors of most previous works [42–45] and was once confirmed with the use of Anchored Hybrid Enrichment technology [9]. Both Rhynchobdellida and Arhynchobdellida formed two monophyletic clades in our trees (Figure 2 and Figure S1 of Supplemental data). The latter order was split into distinct clades of jawed and jawless species, except for *Erpobdella octoculata* KC688270, which clustered as a member of the jawed leech separately from its supposed relatives. As a result of implementing our new approach, *E. octoculata* KC688270 was assigned to the genus *Whitmania* (Figures 3 and 4).

Simultaneous analysis of tree topology, genetic distances and divergence times allowed us to address some clear inconsistencies in taxonomic assignments within Clitellata. Some of them, like *Amynthas-Metaphire* synonymy are supported by literature sources [39,40]. Others, such as *Whitmania* assignment to the family Hirudinidae and dissolution of the genera *Codonobdella* and *Baicaloclepsis* clearly go against conventional systematics and require more careful investigation.

The paraphyly of the families Haemopidae and Hirudinidae was reported previously, albeit with a different topology of the phylogenetic tree [46]. The family Haemopidae itself was first introduced by Richardson [47]; however, his version of systematics was criticized as premature and based on an overestimation of character significance [15]. Nevertheless, species of this family have some hard-to-ignore distinctions from species in the family Hirudinidae in both anatomy and ecology. Based on our analysis, the value of genetic divergence between these families is well within the boundaries of intra-family range, which raises questions about the significance of morphological and ecological criteria and, hence, the validity of Haemopidae (Figures 3 and 4).

According to modern taxonomy, the species *Baicaloclepsis echinulata* and *Baicaloclepsis grubei* belong to the subfamily Toricinae, whereas the genus *Glossiphonia* belongs to the subfamily Glossiphoniinae. Initially, these subfamilies were separated based on the shape of their annuli and their geographic range [48]. However, both the raw genetic distances and divergence times of the *Baicaloclepsis* species group from the members of the genus *Glossiphonia* do not support this split (Figure 4) suggesting the dissolution of the genus *Baicaloclepsis*. Nevertheless, this hypothesis needs further verification, which would require obtaining molecular data on a broader composition of *Glossiphonia* species. The situation is similar for the species of piscine leeches. To resolve the issue of the genus *Codonobdella* validity, we need additional information about the genomes of various piscine leeches outside the genera *Codonobdella* and *Piscicola*.

The high level of genetic distances and divergence times within the group of soil oligochaetes, *Eisenia nordenskioldi* strongly indicates the presence of a cryptic complex consisting of several species (Figures 3 and 4), confirming the results of previous studies (e.g., [49]). On the contrary, within aquatic oligochaetes, the family Naididae has grown excessively due to the recent merge of the former Tubificidae [50], which our results do not support, because their assignment to Naididae creates a cluster of outlier family distances above the intra-order level (Figure 3).

This study mainly aimed to determine the validity of splitting the order Acanthobdellida into two separate families: Acanthobdellidae and Paracanthobdellidae. The degree of their divergence by morphological criteria is more reminiscent of that of a sister species of the same genus. They possess minimal differences in body length and in the shape of their chaetae. The values of both raw and model-adjusted genetic distances between two species of the order Acanthobdellida appeared to be near the lower end of the intra-genus distances distribution, far apart from intra-order distances. In adjusted systematics, some degree of overlap exists between the upper values of intra-genus and lower values of intra-family distances. In summary, the degree of genetic and evolutionary divergence between *A. peledina* and *P. livanowi* indicates that there are no reasons for the formation of a separate family. Based on this fact, we argue that Epstein's original description of this species as *Acanthobdella livanowi* is more realistic [13]. Regarding the validity of the subclass Acanthobdellea, the values of distances at high taxonomic levels are severely overlapped, so its independence from the subclass Hirudinea can neither be reliably affirmed nor denied with this approach. Either way, the distances that separate Acanthobdellea and Hirudinea match more closely with those of different subclasses in the same class. Transferring samples of the order Acanthobdellida into the subclass Hirudinea leads to the formation of the cluster of intra-subclass distances above their upper quartile.

Some issues remain unaddressed. The divergence of the *Whitmania laevis* KC688269 and KM655839 genomes has a value similar to lower values of different species within the same genus, which creates an outlier value of intraspecies distances. However, it is not clear whether these samples should be split into two species, and whether any of them belong to the species *Whitmania laevis*. Unidentified samples of the macrophagous leeches *Erpobdellidae* sp. MT671489 and *Erpobdella* sp. MW435182 are rather distant from one another and from other samples of the genus *Erpobdella*, and create outlier values of the genus distance above the family level. A distinct cluster of intra-order distances above their upper quartile is formed from the distances between jawed and jawless groups of the order Archynchobdellida, which raises the question of whether or not they should be considered one order.

There are, however, pitfalls in our approach. Being relic organisms with archaic characteristics might imply a slower evolutionary rate for the order Acanthobdellida. In ultrametric phylogenies, this, in turn, means that the time scale in this exact branch may be distorted by low values having more "weight" than in other groups. Direct extrapolation of genetic distance values and divergence times from other taxa may be flawed from the start. Additionally, adjusting the samples' taxonomic position in according to their genetic distances and divergence times only makes the dataset a reflection of itself rather than of real evolution and taxonomy, indicating the need for more careful morphological and ecological investigations of all the mentioned taxa. Nevertheless, this might be the best solution we have, while universal standardized criteria are lacking.

## 5. Conclusions

The results obtained allow us to conclude that the proposed method accurately assigns the samples of a particular group of organisms to their respective taxa at different levels (species, genera, families, etc.) by assessing and extrapolating the degree of genetic divergence between samples. Approbation of a new approach to the delimitation of taxa within Clitellata based on complete mitochondrion genome data revealed some inconsistencies in

the existing classification of the group, the fixing of which proposes the following changes in the current systematics of the group:

1.  Transfer *Paracanthobdella livanowi* into the genus *Acanthobdella*, with dissolution of both the genus *Paracanthobdella* and the family Paracanthobdellidae.
2.  Members of the genus *Baicaloclepis* should be synonymized with the genus *Glossiphonia*.
3.  Though the *Piscicola* and *Codonobdella* samples show little genetic divergence, there are significant morphological distinctions between these genera. Because the Piscicolidae samples are underrepresented in the studied dataset, careful reinvestigation of their position is required.
4.  Abolish the family Haemopidae by moving its representatives to the family Hirudinidae. Additionally, more research is required with the extended set of taxa, including the poorly described *Whitmania*.
5.  With growing evidence, it has becomes obvious that oligochaetes of the genera *Amynthas* and *Metaphire* should be considered one genus. Since the former genus was described more than a century prior to the latter, the Code of Zoological Nomenclature retains priority for the genus *Amynthas*.
6.  Regarding the splitting of the *Eisenia nordenskioldi* samples into different species, nothing can be said with confidence based solely on the genetic divergence. More careful morphological investigation is required.
7.  According to our data, the decision to combine Naididae with Tubificidae [50,51] was very hasty and unjustified, since the genetic distances within the Naididae family (in its current state) correspond to the inter-family level. Therefore, the family Tubificidae must be restored.

**Supplementary Materials:** The following supporting information can be downloaded at: https://www.mdpi.com/article/10.3390/d14110948/s1, Table S1: Information on the currently accepted taxonomy of Annelida samples in the analyzed dataset; Table S2: Information on the taxonomy of Annelida samples changed in this work to match genetic and phylogenetic distances in the analyzed dataset; Figure S1: ML tree based on complete mitochondrial genome alignment of 96 annelid specimens, including 88 clitellates and 8 polychaetes used as outgroup.; Figure S2: Distribution of p-distances (a), tree-based distances (b), and dependence of non-ultrametric tree-based distances from p-distances (c) by taxonomic levels according to current sample taxonomy; Figure S3: Distribution of p-distances (a), tree-based distances (b), and dependence of non-ultrametric tree-based distances from p-distances (c) by taxonomic levels according to adjusted sample taxonomy.

**Author Contributions:** Conceptualization, I.K.; methodology, A.B.; software, Y.B. and A.B.; validation, I.K.; formal analysis, Y.B.; investigation, A.B.; resources, I.K.; data curation, I.K.; writing—original draft preparation, A.B.; writing—review and editing, I.K.; visualization, Y.B. and A.B.; supervision, I.K.; project administration, I.K.; funding acquisition, I.K. All authors have read and agreed to the published version of the manuscript.

**Funding:** This research was funded by the Ministry of Education and Science of the Russian Federation (state projects nos. 0279-2021-0008, 0279-2021-0010, and 0279-2021-0011) and the Russian Foundation for Basic Research (RFBR) (grant nos.: 17-29-05097, 19-34-50072, and 19-34-90011). The APC was funded by Irina Kaygorodova.

**Institutional Review Board Statement:** Not applicable.

**Data Availability Statement:** The R script applied to generate the images, and the table data of the current and adjusted sample taxonomies used as references for the script, are available on the FigShare platform (https://figshare.com/s/00a21013944f83039a43, accessed on 1 November 2022). Unpublished figures produced for this research are available in the Supplementary Materials.

**Acknowledgments:** The authors are grateful to their colleagues Piotr Swiatek (University of Silesia in Katowice, Poland), Stanisław Cios (Ministry for Foreign Affairs, Poland), Elena Dzyuba, Natalia Sorokovikova and Valery Chernykh (Limnological Institute, Russia), Maria Baturina (Komi Scientific Centre, Russia), Vitaly Samusenok (Irkutsk State University, Russia), Vitaly Koryagin (Koryak

community, Russia), and Viktor Tarakanov (Tungus community, Russia) for their contribution to the collection of acanthobdellids.

**Conflicts of Interest:** The authors declare no conflict of interest.

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
