# Peer review of "Genome-Based Taxa Delimitation (GBTD): A New Approach"

_diversity, doi:10.3390/d14110948_

Round 1

Reviewer 1 Report

I found this manuscript interesting that has important values. Identification and classification of taxa based on only morphology is often diffcult and misleading in various groups of organisms. Applying DNA-based methods offers a promising solution. A great value in this study that the authors investigated not only a single gene but the complete mitochondrion genome.

However, the method is not fully described. If I understand well, the authors used sequences from species of various genera, families and orders to determine a range of thresholds for separating taxa at different levels and the method is tested on Acanthobdella peledina and Paracanthobdella livanowi if they belongs to the same genus or the same family. I found this approach useful instead of applying a specific threshold for all lineages that could evolve at different rates.

To achieve this, raw and model-adjusted genetic distances were calculated. However, it is not described in the manuscript how the calculations were made. I undestand that calculations were performed in R environment, but what packages and functions were used? Does raw genetic distance mean the number of different sites divided by the number of all of the compared sites expressed in percentage? The name p-distance is written only in the figure captions, however, it should be also used in the text. I assume that genetic distance data at higher taxonomic level did not contain distance data at lower level (e.g. intra-genus distances did not involve intraspecific distances). Is that right?

It would be useful if the authors could made the script available, for example, as a supplementary material.

Besides, I found the using of the term “sample” a bit confusing. If the authours consider that their method would be later applied to other groups of organisms I suggest using “sequence” or “specimen” instead of “sample”. In groups like diatoms where DNA-based methods (e.g. metabarcoding) have been also introduced the sample refers to a community of several taxa not a single specimen.

I suggest a major revision with a more detailed description of the method.

Minor comments:

Page 1, line 12: please change „genetic sequences” to „DNA sequences”

Page 1, line 15: please change “taxonomic position of the samples” to “taxonomic position of taxa”

Page 1, line 35: what do mean under “very concept”? Maybe something is missing here?

Page 1, lines 45-46: I would argue with that statement “morphological criteria solely do not provide sufficient resolution for modern research”, I think it depends on taxonomic group and the aim of the research. What do you mean by “modern research”? Please specify.

Page 2, line 51: Metabarcoding was applied to various organisms. Please specify what group of organisms 2-3% threshold was established for.

Page 2, line 90: Please insert citation after Epstein’s name

Page 3, line 109: Please specify what functions were used in the script for distance calculation and iteration.

Page 3, line 114: Is “sample pairs” correct? Based on Figure 1, I suspect it would have be “taxa pairs”.

Page 3, line 116: I suggest changing “sample ID” to “sequence ID”.

Page 4, line 143: What additional measures were taken?

Page 5, line 155: Maybe “(a)” was left in the text by mistake. If yes, please delete it.

Page 5, line 195: Please write out the substitution model. What does X means?

Page 5, line 205: This section is Results not Results and discussion. Please correct the title.

Page 7, Fig. 2: Does the colour bar represent posterior probability? Please indicate it in the figure caption.

Page 7, line 262: I suggest changing “sample” to “sequence”.

Page 7, line 268: manillensis should be written in italic

Page 7, line 269: I think “specimen misidentification” should be better than “sample misidentification”.

Page 8, line 286 and page 9, line 316: I think Tibificidae is a typo, please correct it.

Page 8, line 294: geometra should be written in italic

Page 9, line 296: I suggest changing “samples” to “specimens” or “sequences”.

Page 9, line 302: with exception of 1

Page 9, line 303: The approach applied by the authors is not DNA-barcoding because it is based on not only a short “barcode” region but the whole mitochondrion genome. I suggest changing “DNA-barcoding approach” to “DNA-based approach” or “mitochondrion genome-based approach”

Page 10, line 333: The name of genus of A. livanowi should be written in full.

Page 10, line 358: I think “are” is unnecessary, please delete it.

Page 10, line 370: I suggest changing “broader composition” to “broader sampling”.

Page 11, line 406: I suggest changing “samples” to “sequences”.

In the Discussion more citations should be inserted to the following parts:

- Page 10, lines 336: for “Epstein”

- Page 10, lines 350-351: for “literature sources”

- Page 10, line 352: for “conventional systematics”

- Page 10, line 363: for “modern taxonomy”

- Page 10, lines 384-385: for the sentence “They possess minimal differences in body length and the shape of chetae”

Author Response

Thank you for a high estimate of the GBDT approach to delimitation of different taxonomic levels, which we presented in this article.

Your valuable comments helped us to improve our manuscript.

Using a new approach, we did not aim to determine specific threshold values, but rather estimate the distributions of genetic distances characteristic of certain taxonomic levels and determine which distributions specific values belong to. Perhaps the method is not sufficiently explained. We added more description into the section 2.1.

In the section 2.1, we added information on packages and methods were used.

Regarding your question about genetic distances, the distances between samples are pairwise. If samples share the same family, but are of different genera, the distance between them will be plotted just once in the «family» distribution, not higher or lower.

The GBTD script and all input files used for this work are accessible online. There is reference in the Section “Data Availability Statement”.

All Minor comments were taking into account.

Reviewer 2 Report

The authors present an impressive dataset of new and archival annelid mitochondrial DNA sequences, and thereafter describe and execute a combinatorial set of computational analyses to decipher taxonomic divisions. The study is timely in that species delimitation is currently a topic of considerable debate, and the approach taken in this study offers a set of tools to consider in the broader conversation. The text is generally well-articulated and the authors have been careful not to overextend their interpretations, even pointing out potential pitfalls. Appropriate recommendations are given in the Conclusion based on the completed analysis.

The manuscript can be improved by a more careful comparison between "current" and "adjusted" sample taxonomy. For example, the fusion of Figs. 3 and 4, with side-by-side alignments of the two approaches along with highlighting of features that distinguish them, will help the reader understand the advantages of the new approach. This should be coupled with corresponding text, in the legend and Results section. Also, written text in these Figures is too small to read on a printed version. 

Additionally, it is not clear why the authors repeat their affiliation three times on the Title page? Section "3. Results and discussion" should be "3. Results". Italicize "Hirudo nipponia" Line 267. In general, Results should be in past tense throughout. Line 358, delete "are". Line 449, provide supporting references after 1st sentence of item 7.

Finally, I suggest to assign a formal name to this approach as opposed to referencing it as a "new method". Ideally, this could be compiled into software that can be accessed and applied to other taxonomic studies. 

Author Response

Dear Reviewer,

Thank you for your feedback and high estimate of our work. Your comments really helped us to improve our manuscript.

Regarding your comment that results of “current” and “adjusted” sample taxonomy inference should be compared side-by-side, you are right. Differences between "current" and "adjusted" sample taxonomies would be easier to see in one Figure. However, Figures 3 and 4 are too large to combine into one. And, as you correctly pointed out later, the text in these figures is too small to be read even now, when these figures are separated.

We've resized these Figures to a larger format and increased the font size to improve the visibility of the text on them.

Regarding triple affiliation repeat, we have followed the requirements of the submission system and hope that this will be changed by copy editor.

We have also fixed the typos you mentioned and gave our approach a name GBTD (Genome-Based Taxa Delimitation).

Round 2

Reviewer 1 Report

See attached file.

Author Response

A few typos you found have been corrected. Thank you for your vigilance and responsible attitude to the review.